# Mechanism Underlying Metformin Action and Its Potential to Reduce Gastric Cancer Risk

**DOI:** 10.3390/ijms232214163

**Published:** 2022-11-16

**Authors:** Wen-Hsi Lan, Ting-Yu Lin, Jia-Ai Yeh, Chun-Lung Feng, Jun-Te Hsu, Hwai-Jeng Lin, Chia-Jung Kuo, Chih-Ho Lai

**Affiliations:** 1School of Medicine, College of Medicine, Chang Gung University, Taoyuan 33302, Taiwan; 2Department of Microbiology and Immunology, Graduate Institute of Biomedical Sciences, Chang Gung University, Taoyuan 33302, Taiwan; 3Division of Gastroenterology and Hepatology, Department of Internal Medicine, China Medical University Hsinchu Hospital, Hsinchu 30272, Taiwan; 4Department of Internal Medicine, Department of Medical Research, School of Medicine, China Medical University and Hospital, Taichung 40402, Taiwan; 5Department of General Surgery, Chang Gung Memorial Hospital at Linkou, Taoyuan 33305, Taiwan; 6Division of Gastroenterology and Hepatology, Department of Internal Medicine, Shuang-Ho Hospital, New Taipei 23562, Taiwan; 7Division of Gastroenterology and Hepatology, Department of Internal Medicine, School of Medicine, College of Medicine, Taipei Medical University, Taipei 11031, Taiwan; 8Department of Gastroenterology and Hepatology, Chang Gung Memorial Hospital at Linkou, Taoyuan 33305, Taiwan; 9Chang Gung Microbiota Therapy Center, Chang Gung Memorial Hospital at Linkou, Taoyuan 33305, Taiwan; 10Molecular Infectious Disease Research Center, Department of Pediatrics, Chang Gung Memorial Hospital at Linkou, Taoyuan 33305, Taiwan; 11Department of Nursing, Asia University, Taichung 41354, Taiwan

**Keywords:** metformin, gastric cancer, diabetes mellitus, reactive oxygen species, microbiota

## Abstract

Diabetes mellitus is associated with a high risk of developing gastric cancer (GC). Metformin, which is conventionally used to treat type 2 diabetes, induces AMP-activated protein kinase signaling and suppresses gluconeogenesis. Recent studies have reported that metformin is associated with beneficial effects in cancer prevention and treatment owing to its anti-tumor effects. This makes metformin a potential medication for GC therapy. However, contradicting reports have emerged regarding the efficacy of metformin in reducing the risk of GC. This review summarizes the impact of metformin on mitigating GC risk by analyzing clinical databases. The mechanism underlying the anti-tumor effect of metformin on GC is also discussed.

## 1. Introduction

Gastric cancer (GC) is a global health issue with over a million cases and 700,000 deaths worldwide by 2020 according to GLOBOCAN [1]. Risk factors for the disease include age, sex, genetics, tobacco smoking, and high nitrate or nitrite diet [2]. Nevertheless, *Helicobacter pylori* (*H. pylori*) infection remains the primary risk factor for the onset of GC in more than 89% of cases worldwide [3]. *H. pylori* is usually acquired at a young age; if left untreated it persists lifelong. Advances in adjuvant and neoadjuvant chemotherapy, targeted therapy, and immunotherapy have also improved the outcomes of advanced GC. Despite the decline in the incidence and mortality rates of GC, various treatment options need to be explored as the currently available treatments are flawed. Surgical resection of gastric tumor is either invasive or unviable if the diagnosis is made at a late stage when the tumor has metastasized—as is often the case [4]. In such cases, chemoradiation and cytotoxic therapy are the only options available; however, they are associated with adverse effects [5].

Similar to GC, diabetes mellitus (DM) is an increasingly prevalent global disease [6]. Previous studies have demonstrated that DM predisposes patients to a higher risk of cardiovascular disease and cancer development [7]. The association between GC and DM remains unclear because contradictory results have been reported. A previous study reported that DM increases the incidence of GC by 67% [8]. DM is correlated with higher mortality rates in patients with GC [7]. Several mechanisms through which DM promotes tumor development such as hyperglycemia-induced DNA damage, increased production of reactive oxygen species (ROS), and stimulation of cell proliferation and angiogenesis due to altered glucose metabolism have been reported [7,8].

Metformin, a guanidine derivative, is commonly used for the treatment of type 2 diabetes (T2D) [9]. The inhibition of mitochondrial complex I and activation of AMP-activated protein kinase (AMPK) enables metformin to lower glucose levels with minimal adverse effects [10]. Its anti-tumor properties include decreasing ROS generation and inducing autophagy [10]. Additionally, metformin restores the microbiome diversity, which is closely linked to *H. pylori* colonization and GC progression [9]. Thus, metformin needs to be investigated as a potential intervention for GC, and its underlying mechanism in treating GC needs to be elucidated.

## 2. Metformin Mechanism of Action and Diabetes Mellitus

Although metformin is the drug of choice to initiate diabetes mellitus treatment, how it lowers hepatic glucose production to benefit diabetes is still not fully understood [11]. Metformin inhibits mitochondrial respiratory chain complex 1, leading to the suppression of gluconeogenesis, followed by alteration of hepatic energy and AMPK activation [12]. Reduced electron transport chain activity decreases ATP generation [13] as well as the ATP/ADP and ATP/AMP ratios. AMPK is a cellular energy sensor that is directly phosphorylated and activated by liver kinase B1 (LKB1) or calcium/calmodulin-dependent protein kinase 2 [14,15]. Additionally, AMPK was activated by an increase in the ADP/ATP and AMP/ATP ratios (Figure 1). Although the energy stress mechanism is assumed to be canonical, evidence of other ATP level-independent mechanisms is accumulating. Moreover, higher AMP levels in cells may also inhibit the key gluconeogenesis enzyme, fructose 1,6-phosphate (FBP1).

Shaw et al. indicated that in high-fat diet-fed mice, the deletion of LKB1 in the liver inhibited AMPK activity and led to hyperglycemia [16]. This study also showed that the LKB1-AMPK signaling pathway might repress the gluconeogenic transcriptional pathway by the phosphorylation and nuclear exclusion of the CREB-regulated transcription coactivator 2. However, a later study challenged the mechanism of transcriptional inhibition by AMPK activation. The results showed that metformin inhibition of gluconeogenesis was amplified in both LKB1- and AMPK-deficient hepatocytes [17]. Metformin affects the hepatic energy state and changes the NADH/NAD^+^ ratio, inhibiting glucose production.

Furthermore, Miller et al. suggested that metformin acts by suppressing glucagon signaling [12]. They reported that metformin could decrease cyclic 3′,5′-adenosine monophosphate (cAMP) production after glucagon stimulation in hepatocytes, thus blocking glucagon-dependent glucose output from hepatocytes. However, the relevance of glucagon in the mechanism of action of metformin is doubtful because of the very high dose used in the study (500 mg/kg).

Metformin-induced activation of AMPK in hepatocytes reduces acetyl-CoA carboxylase (ACC) activity and suppresses lipogenic transcription factor, sterol regulatory element-binding transcription factor 1 (SREBF1) expression, which may be linked to a reduced fatty liver [18]. Phosphorylation of ACC by AMPK reduces lipogenesis and promotes hepatic mitochondrial oxidation of fatty acids, resulting in a lipid-lowering and insulin-sensitizing effect exerted by metformin [19]. Therefore, AMPK is essential for metformin for the inhibition of glucose production by metformin in the hepatocytes.

Metformin has been demonstrated to inhibit small intestinal glucose absorption in T2D patients [20]. This action may be relevant to enhanced secretion of glucagon-like peptide 1 (GLP1) and regulation of gut microbiota [21]. Metformin inhibits glucose absorption in the proximal small intestine and increases glucose utilization in intestinal enterocytes via the recruitment of glucose transporter 2 to the apical membrane [22]. Metformin also, directly and indirectly, stimulates the secretion of GLP1 from L cells in the distal small intestinal mucosa. GLP1 is an incretin that acts through the gut–brain–liver neuronal axis to reduce hepatic glucose production. Furthermore, GLP1 enhanced glucose-stimulated insulin secretion and inhibited pancreatic glucagon secretion [11]. Through the reduction in ileal absorption, metformin increases the intestinal bile acid pool, which contributes to cholesterol homeostasis, glucose homeostasis, and elevated GLP1 levels [23,24]. In addition, metformin can slow gastric emptying, which plays an important role in the glucose-lowering effect after a meal [25]. It is well known that bile acids are key regulators of glucose metabolism. Metformin reduces gut bile acid resorption substantially and then exerts its glucose-lowering activity [26].

## 3. Clinical Use of Metformin

### 3.1. Metformin Use Reduces GC Risk in Patients with Diabetes

Diabetes and GC are often associated with each other owing to shared risk factors, such as hyperglycemia, hyperinsulinemia, insulin resistance, high salt intake, and cigarette smoking [27]. GC diagnosis leads to a poorer prognosis in patients with diabetes than those without diabetes [27]. In addition to diabetes, *H. pylori* infection is a critical risk factor for GC. Previous research has shown that patients with diabetes exhibit a higher infection rate and lower *H. pylori* eradication rate [28], further strengthening the link between diabetes and GC. Metformin is conventionally used to treat T2D; however, its antitumor effects make it a potential drug for treating GC. Alleviation of the adverse effects of diabetes reduces risk factors for GC. Thus, controlling diabetes may suppress the progression of GC.

### 3.2. Metformin and GC Prevention

The efficacy of metformin in preventing GC was evaluated by analyzing the incidence of GC between the ever-users and non-users of metformin in T2D patients based on the National Taiwan Insurance database. The statistical results showed that the incidence rate was 0.26% and 0.55%, respectively, with a hazard ratio of 0.448, suggesting a significantly lower risk among ever-users [29] (Table 1). Additionally, the data presented a time-dependent factor. The patients prescribed metformin for more than two years exhibited a significant reduction in GC risk, and long-term treatment with metformin lowered the morbidity of GC compared with diabetes treatment without metformin. Different models of metformin use based on variables, such as combined use with other antidiabetic drugs, statins, and *H. pylori* infection, were studied to investigate the joint effect of other drugs and potential risk factors. The results showed that metformin significantly reduced the risk of GC, independent of the presence of *H. pylori* infection and the use of other antidiabetic drugs [29]. *H. pylori* was validated as a risk factor for GC by a prominent increase in the hazard ratio in the model of metformin use with *H. pylori* infection compared with that in the other models.

A similar trend was detected for GC inhibition by metformin on analysis of the statistics pertaining to patients with T2D in the Korean National Health Insurance Database. Metformin use for more than three years was associated with a significant reduction in GC risk by 43% among those who used metformin compared with those who did not [30]. The duration of metformin use was positively correlated with a reduced risk of GC; however, the difference was not significant for insulin users. In fact, the risk of GC doubled in insulin users compared with that in non-users, independent of metformin use. According to some previous population-based observational studies, insulin may be related to an increase in cancer risk owing to the increased expression of IGFR and the corresponding signaling pathway, leading to stimulation of cell proliferation [41]. Another possible explanation for the association between insulin and GC risk may be *H. pylori* infection. The statistics from the Korean National Health Insurance Database do not present information on whether the patients had *H. pylori* infection.

As insulin can be used to eradicate *H. pylori*, it is often prescribed to patients with a more serious case of *H. pylori* infection, which in itself is a risk factor for GC. The association between *H. pylori* infection and glycemic control in those with GC and had been prescribed metformin was analyzed using the Clinical Data Analysis and Reporting System (CDARS) of the Hong Kong Hospital Authority. In patients receiving *H. pylori* eradication therapy, metformin use exhibited a 51% reduction in GC risk, whereas the sole use of *H. pylori* eradication therapy reduced GC risk by only 40%. Metformin demonstrated remarkable efficacy in reducing risk and was strongly associated with dose and duration gradients [34]. These results also indicate another potential therapeutic approach to preventing GC by combining *H. pylori* eradication and metformin prescription. Interestingly, the results showed that the efficacy of metformin was independent of HbA1c levels in *H. pylori*-eradicated patients [34], indicating that glycemic control has little effect on metformin treatment.

As discussed earlier, hyperinsulinemia is related to cancer risk, whereas metformin is an insulin-sensitizing drug. An Italian population-based cohort study found that metformin reduced the incidence of most types of digestive cancers, including GC. The study analyzed different types of drugs for diabetes treatment; however, metformin was the only medication that showed a significant association with a decreased risk for most types of digestive cancers [31]. However, the mechanism underlying the anti-tumor action of metformin remains elusive. Additional studies are required to establish a biochemical model for analyzing the molecular mechanism underlying GC prevention by metformin.

However, some studies that used different databases did not find a significant relationship between metformin and the reduction in GC risk. Two cohort studies were conducted using the Swedish Prescribed Drugs and Health Cohort (SPREDH). One of the studies included individuals with diabetes who were medicated with anti-diabetic drugs other than metformin. The other study included common-medication users, setting metformin users as the exposed group [37]. The results from both cohort studies showed that metformin did not reduce GC risk.

Another study conducted in the Netherlands that investigated the association between metformin and GC also arrived at a similar conclusion. The Netherlands Cancer Registry (NCR) PHARMO database was analyzed, and metformin users were compared with users of other non-insulin antidiabetic drugs (NIADs). The risk of GC was not significantly different between metformin users and other NIADs users [33]. However, the sample size of the GC arm was only 53 patients, which may not be sufficiently representative.

### 3.3. Metformin Use in GC Treatment

In addition to the potential use of metformin in GC prevention, another potential clinical application could be as a drug to reduce mortality. A study based at Chang Gung Memorial Hospital in Taiwan analyzed patients with GC and DM post-gastrectomy. The results showed that among patients with stage III GC, metformin users had significantly prolonged cancer-specific survival; the results were not significant in stages I and II GC [38]. Apart from improving survival, metformin also decreased the risk of tumor recurrence, confirming metformin as a potential adjuvant drug in the treatment of gastrectomy. A Korean study in GC patients with diabetes who had undergone gastrectomy showed that medication with metformin for longer than 6 months significantly decreased the risk of recurrence, cancer-specific mortality, and all-cause mortality. In addition, survival rates were comparable to those of patients without diabetes [32]. According to a previous study, metformin is especially beneficial for GC patients with diabetes by successfully reducing mortality.

Compared with studies undertaken in Taiwan and Korea, the results of a Belgian population-based study showed that despite reduced all-cause mortality in GC patients with diabetes, cancer-specific mortality did not improve [35]. In addition, the relationship between metformin and survival was dose independent. The possible reason for the reduced mortality needs to be clarified by investigating the mechanism underlying the anti-tumor effect of metformin.

Although the results of these studies differ regarding the efficacy of metformin in reducing mortality in GC patients, the consensus is that metformin is efficacious in GC treatment. However, the statistics from the Lithuanian Cancer Registry and the National Health Insurance Fund Database estimating the survival of patients with GC while taking various medications for T2D did not show a significant association between the reduction in GC mortality and treatment with metformin. In other studies, insulin was found to be related to increased GC risk. However, the mortality rate of GC patients is not different from that of other anti-hyperglycemic drugs or non-diabetic patients [36]. The detailed mechanism that leads to different tendencies in morbidity and mortality in patients with GC needs further exploration.

## 4. Repositioning (Repurposing) of Metformin for Cancer Inhibition

In addition to its effect on gluconeogenesis and insulin sensitivity, LKB1 functions as a tumor suppressor [42]. The anti-tumor effect exerted by metformin may be attributed to its interaction with LKB1, which is most pronounced in the regulation of cell growth. Metformin inhibits the proliferation of several human GC cell lines including MKN1, MKN45, and MKN74 [43]. High concentrations of metformin strongly inhibited cancer cell growth in dose- and time-dependent manners. Metformin inhibits cancer cell proliferation by decreasing the expression of cyclins, such as cyclin-dependent kinase (Cdk) 4, Cdk6, cyclin E, and Cdk2, and reducing the level of phosphorylated retinoblastoma protein. Loss of cyclin D1 prevents cells from transitioning from the G0 to G1 phase, suppressing normal cell cycle progression [43]. The results of in vivo studies showed that metformin inhibits cell proliferation. In experiments where MKN74 cells (a gastric adenocarcinoma cell line) and metformin were injected into nude mice [43], a decrease in tumor growth was observed as a result of G1 cell cycle arrest. Additionally, the tumor volume dramatically decreased as the metformin dosage increased. Metformin also reduced the expression of phosphorylated epidermal growth factor receptor (p-EGFR) and phosphorylated insulin-like growth factor-1 receptor (p-IGF-1R) in vitro and in vivo [43]. In addition, metformin regulates the mTOR pathway and decreases the expression of surviving [44], which is involved in apoptosis inhibition and cell cycle regulation [45]. Furthermore, metformin downregulates hepatocyte nuclear factor 4 alpha (HNF4α) by activating AMPKα, leading to cyclin downregulation, cell cycle arrest, and tumor growth inhibition [46].

EGFR, a receptor tyrosine kinase of the ErbB family, is actively expressed in a variety of solid tumors [47]. IGF-1R has long been recognized for its role in tumorigenesis [48]. Both EGFR and IGF-1R are involved in tumor cell proliferation. Metformin inhibits cancer cell proliferation by reducing the expression of EGFR and blocking signaling through IGF-1R [49,50,51]. Treatment with metformin modulates miRNA levels in diabetic patients and reduces cyclin levels in several cancer cell lines. For example, in the SGC-7901 GC cells, metformin increases the expression of miR-107, which inhibits tumor growth and invasion, induces apoptosis, and is associated with cell cycle arrest [52]. Metformin along with miR-365 promotes apoptosis in GC cells [53]. Additionally, the epithelial-to-mesenchymal transition and self-renewal properties induced in cancer stem cells by the Wnt/β-catenin pathway are reversed by metformin in GC [54,55]. Long noncoding RNAs have been implicated to be possibly oncogenic in several cancers. A higher expression level of Loc100506691 RNA was associated with poor survival in patients with GC. However, metformin was shown to decrease the Loc100506691 RNA level in GC cells [56] (Table 2).

## 5. Metformin Use Alters Microbiota Composition

The gut microbiota contributes to enteric protection against pathogens, metabolic regulation of nutrients and drugs [60], and the maintenance of the intestinal barrier [61]. Many diseases, such as diabetes and obesity, are associated with alterations in the composition of the gut microbiota, and *H. pylori* infections are known to reduce gastric bacterial diversity [60]. Gut microbiota contributes to carcinogenesis through metabolite production. For instance, certain bacteria produce butyrate, which promotes the proliferation of abnormal epithelial cells [62]. N-nitroso compounds (NOCs) produced by *E. coli*, *Lactobacillus*, and *Nitrospirae* [63] promote mutagenesis, angiogenesis, of proto-oncogene expression, and apoptosis inhibition [64].

One of the anti-tumorigenic properties exhibited by metformin has been the restoration of gastric bacterial diversity by directly inhibiting bacterial growth and modifying gastrointestinal physiology [60]. For example, metformin enhances intestinal bile acid secretion, glucose release [60], and lipopolysaccharide production [65]. These changes in the gastric microenvironment consequently affect the bacterial communities.

In murine models, metformin treatment has increased the relative abundance of many short-chain fatty acid (SCFA)-producing bacteria, such as *Bacteroides* and *Butynococcus* [9]. Metformin also increases the abundance of probiotic organisms such as *Akkermansia* spp. [9], *Faecalibaculum*, and *Bifidobacterium* [66]. *Akkermansia* spp. attenuate cancer development by regulating glucose metabolism [67], while *Bifidobacterium* spp. produce metabolites that create an anti-tumorigenic environment by lowering gastric pH [68] and inflammatory responses [60]. Metformin decreased the abundance of the tumorigenic bacteria *Lactobacillus* and *Clostridium,* the latter of which contributes to gastric tumorigenesis by producing carcinogenic factors [69]. It also effectively decreased the relative abundance of *Firmicutes* in *H. pylori*-infected murine models [60].

## 6. Metformin and Adverse Effects

Metformin’s most common adverse effects are gastrointestinal symptoms, including nausea, diarrhea, vomiting, and intestinal discomfort [70]. Vitamin B12 is involved in red blood cell formation, and long-term metformin use has been reported to cause low vitamin B12 levels, leading to anemia and peripheral neuropathy [71]. Because metformin reduces hepatic gluconeogenesis and glucose uptake, metformin overdose can result in severe hypoglycemia [72]. Lactic acidosis and acute pancreatitis have been reported in patients with renal insufficiency because unmetabolized metformin cannot be properly excreted. Metformin inhibits mitochondrial respiration in the liver, leading to increased plasma lactate concentrations [73,74]. Although the incidence of metformin-induced lactic acidosis is infrequent, it is associated with high mortality and should be considered before the prescription. Hepatotoxicity is a rare adverse effect reported in metformin-treated patients [75].

## 7. Conclusions and Perspectives

Metformin has been widely prescribed as a T2D drug for decades, and its effects and mechanisms have been extensively studied. Mounting evidence has revealed its antitumor properties and potential as a GC prevention drug and medication. Apart from its biochemical effects, metformin also contributes to GC treatment by altering the gut microbiome. In this study, we have summarized the mechanisms by which metformin inhibits cancer cells and comprehensively discussed the efficacy of metformin in GC prevention by analyzing databases from different countries. The adverse effects of metformin have also been discussed, and it is considered a safe drug with low risks.

The repositioning or repurposing of metformin for the inhibition of several types of cancer has been reported elsewhere. Although the mechanism underlying the antitumor effect of metformin on GC has been investigated in cell-based studies, the information available from in vivo studies is currently insufficient. Additionally, analyses based on databases from different countries have yielded conflicting results. The role of metformin in the prevention or treatment of T2D in patients with GC remains controversial. Long-term follow-up studies with large-scale prospective trials on the effects of metformin in GC treatment are warranted to elucidate the precise mechanism by which metformin exerts its antitumor effect.

## Figures and Tables

**Figure 1 ijms-23-14163-f001:**
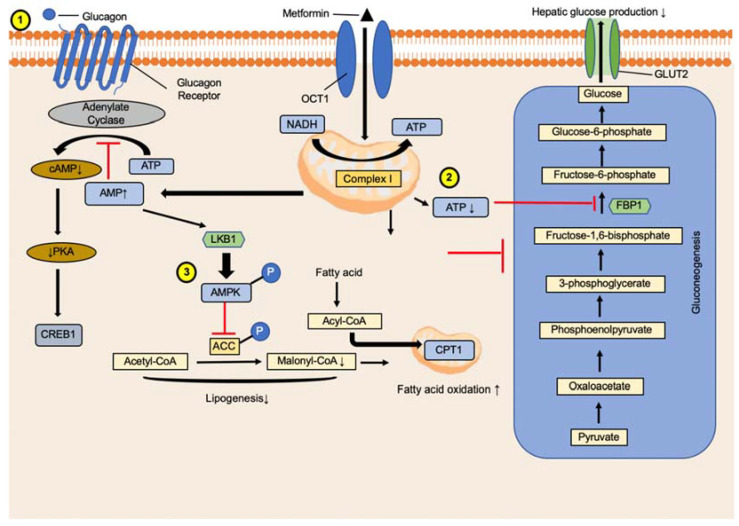
Mechanism deciphering the role of metformin in glucagon signaling. (1) Metformin acts by suppressing glucagon signaling, leading to decreased cAMP production in hepatocytes and lower glucagon-dependent glucose output from hepatocytes. Reduced cAMP synthesis lowers the activity of gluconeogenic gene phosphorylation regulators cAMP responsive element binding protein 1 (CREB1) and triiodide (I3PR). (2) ATP reduction lowers gluconeogenesis and increases AMP levels, which inhibits the key gluconeogenic enzyme fructose-bisphosphatase 1 (FBP1). (3) The acetyl-CoA carboxylase (ACC) was phosphorylated and inhibited malonyl-CoA and acyl-CoA synthesis, increasing acyl-CoA content entering mitochondria and leading to the oxidation of fatty acid.

**Table 1 ijms-23-14163-t001:** The effects of metformin use on GC patients.

Author and Study Year	Study Location	Effect of Statins on GC	Joint Effect of Other Drugs	Presence of *H. pylori*	Reference
Kim, 2014	Korea	Metformin use for >3 years was associated with a significant 43% reduction in GC risk among those who did not use insulin	GC risk might be doubled in insulin users while compared to nonuser, disregarding metformin use	NA ^†^	[30]
Valent, 2015	Italy	Metformin prescription reduced risk of most types of digestive cancer	NA	- ^‡^	[31]
Lee, 2016	Korea	Metformin use for more than 6 months decreases recurrence, cancer-specific mortality, and all-cause mortality in diabetic patients with gastrectomy	NA	NA	[32]
Tseng, 2016	Taiwan	Metformin significantly reduces GC risk, especially for patients prescribed more than 2 years	Independent of insulin, sulfonylurea, meglitinide, acarbose, rosiglitazone, pioglitazone, and statin	+/− ^¶^	[29]
de Jong, 2017	Netherlands	No significant difference between metformin users and other NIADs users	NA	-	[33]
Cheung, 2018	Hong Kong	In *H. pylori*-eradicated patients, metformin use exhibited a 51% reduction in GC risk, and associated with dose- and duration-gradients	Independent with glycemic control	-	[34]
Lacroix, 2018	Belgium	Metformin use reduces all-cause mortality but not for cancer-specific mortality	NA	NA	[35]
Dulskas, 2019	Lithuania	No significant effect on survival in patients prescribed metformin	Insulin	NA	[36]
Zheng, 2019	Sweden	Metformin use does not reduce GC risk in both diabetic population and common-medication population	No significant difference in combination of other anti-diabetic drugs	-	[37]
Chung, 2020	Taiwan	Metformin prolonged cancer-specific survival in GC stage III patients	NA	NA	[38]
Cho, 2021	Korea	Metformin reduced mortality among diabetic and nondiabetic populations with use for ≥547.5 cDDD-days	Aspirin and statin	NA	[39]
MacArthur, 2021	Olmsted County, Minnesota	No protective effect of metformin against early onset GC	NA	+/−	[40]

^†^ Not available; ^‡^ non-*H. pylori* infection; ^¶^ including both *H. pylori* infection and non-*H. pylori* infection.

**Table 2 ijms-23-14163-t002:** Mechanism of metformin in the inhibition of cancer cells.

Author and Study Year	Experiment	Target or Mechanism	Reference
Ryan, 2009	Review strategies to target survivin in cancer cells	Survivin is one of the most cancer-specified proteins and is upregulated in nearly all human tumors	[45]
Kato, 2012	Dose–response of metformin vs. viability and cyclins in MKN1, MKN45, and MKN74 human GC cell lines and tumor growth of inoculated GC cell line in nude mice	Metformin decreases the expression of cyclins thus can inhibit tumor growth by arresting G1 cell-cycle in vivo	[43]
Zhou, 2015	Linc00152 expression and interaction protein identification	Linc00152 promotes proliferation in GC through the EGFR-dependent pathway	[57]
Chang, 2016	Integrative analysis of transcriptomic profiles of tumors and their matched noncancerous samples. The candidate genes were validated by inhibition using RNA interference	AMPKα downregulates HNF4α, resulting in cyclin downregulation, cell cycle arrest, and tumor growth inhibition	[46]
Courtois, 2017	Study the metformin effects on proliferation and tumorigenic properties of primary cell culture from patients and established GC cell lines	Metformin leads to tumor growth delay and decreases cancer stem cell’s renewal ability	[58]
Chen, 2020	Identify potential metformin targets in cancer cells using bioinformatics analysis. Validate targets using miR-107 mimic and metformin treatment	Metformin increases miR-107 expression in the SGC-7901 GC cell line. MiR-107 inhibits tumor growth and invasion, inducing apoptosis and is related to cell-cycle arrest	[52]
Tseng, 2021	Identify metformin targets using microarray data by treating several metformin concentrations. Validate the target genes using gene knockdown experiment	GC cells treated with metformin can decrease the oncogenic long noncoding RNA Loc100506691, while high Loc100506691 expression level is correlated with poor GC patient survival	[56]
Valaee, 2021	Study metformin concentration vs. vimentin expression level in AGS cells. Assess vimentin function by siRNA (vim-siRNA) knockdown method	Vimentin is related to cancer metastasis and upregulated in epithelial-mesenchymal transition. Suppression of vimentin expression may have a beneficial effect on cancer survival. Metformin can be used to suppress vimentin expression	[55]
Zou, 2021	Investigate TGF-β1 expression level and reporter gene activity vs. metformin treatment. The roles of Smad3 were determined by gene knockdown and deleting Smad3 binding sites in the TGF-β1 promoter region	AMPK suppression of TGF-β1 autoinduction is mediated through the inhibition of Smad3 phosphorylation and activation	[59]
Huang, 2022	Assess apoptosis of GC cells and AMPK activity by metformin treatment. Analysis of metformin and miR-365, individually and combined, on the apoptosis of GC cells and the miR-365-PTEN-AMPK axis activity	Metformin can cooperate with miR-365 to promote GC cell apoptosis via the miR-365-PTEN-AMPK axis	[53]

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
