# Peer review of "Mechanism Underlying Metformin Action and Its Potential to Reduce Gastric Cancer Risk"

_ijms, 2022, doi:10.3390/ijms232214163_

Round 1

Reviewer 1 Report

The manuscript is not well structured to reflect the focus outlined in the title.

The effects of metformin on glycaemia were inappropriately summarised. There is increasing recognition of the gastrointestinal tract being key to the actions of metformin, which was completely ignored. The authors stated rather controversial and out of date opinions about the glucose-lowering action of metformin.  

The potential for metformin to reduce the risk of or to treat gastric cancer should be first stated on the basis of clinical data, followed by the mechanisms of action, which should be carefully discussed in different sections. 

The tables and figures included in this manuscript therefore lack clarity or of little relevance to the focus of this manuscript.

Reviewer 2 Report

This review describes the mechanism of metformin and its potential to reduce the risk of gastric cancer.

1.     Please include a section on the side effects of metformin, since despite having several effects that are shown to be very favorable, there are others that will be important to consider.

2.     Figure 1 has no context in the writing, at no time is glucagon discussed with respect to the topic being addressed in general. Additionally, several other proteins and metabolic pathways in the figure are mentioned without context in the manuscript. Please remove figure 1 and place another one with the mechanism of action of metformin according to what is described in the text.

3.     Number 2 is a description of the mechanism of action and not of metformin and diabetes. Please change the title and add another section with metformin and T2DM.

4.     The conclusions are very scarce. There is a lack of depth.

Round 2

Reviewer 1 Report

The authors have made substantial efforts to address my concerns, particularly expanded discussions on the actions of metformin in relation to glycaemic control. The structure of the review now flows much better. Some additional gastrointestinal actions of metformin should be included in the discussion. For example, metformin was known to inhibit small intestinal glucose absorption in patients with type 2 diabetes (PMID: 27761984). This action may be relevant to enhanced secretion of GLP-1 and alteration of gut microbiota. Moreover, metformin slows gastric emptying (PMID: 30370686), which is important to its glucose lowering effect after a meal. The interaction of metformin and bile acids has been summarised in a recent review, which may be referred to (PMID: 31468642).

Reviewer 2 Report

The authors have made the requested modifications; the figure shows the information described in the text regarding the mechanism of action of metformin. They changed the title and added a section accordingly. And conclusions were revised by adding a comprehensive summary.
